# Transcription mediated insulation and interference direct gene cluster expression switches

**Tania Nguyen[1], Harry Fischl[1†‡], Françoise S Howe[1†‡], Ronja Woloszczuk[1†‡], Ana Serra Barros[1§], Zhenyu Xu[2§], David Brown[1], Struan C Murray[1], Simon Haenni[1], James M Halstead[1], Leigh O'Connor[1], Gergana Shipkovenska[1], Lars M Steinmetz[2], Jane Mellor[1]\***

[1]Department of Biochemistry, University of Oxford, Oxford, United Kingdom; [2]Genome Biology Unit, European Molecular Biology Laboratory, Heidelberg, Germany

**Abstract** In yeast, many tandemly arranged genes show peak expression in different phases of the metabolic cycle (YMC) or in different carbon sources, indicative of regulation by a bi-modal switch, but it is not clear how these switches are controlled. Using native elongating transcript analysis (NET-seq), we show that transcription itself is a component of bi-modal switches, facilitating reciprocal expression in gene clusters. *HMS2*, encoding a growth-regulated transcription factor, switches between sense- or antisense-dominant states that also coordinate up- and down-regulation of transcription at neighbouring genes. Engineering *HMS2* reveals alternative mono-, di- or tri-cistronic and antisense transcription units (TUs), using different promoter and terminator combinations, that underlie state-switching. Promoters or terminators are excluded from functional TUs by read-through transcriptional interference, while antisense TUs insulate downstream genes from interference. We propose that the balance of transcriptional insulation and interference at gene clusters facilitates gene expression switches during intracellular and extracellular environmental change.

\*For correspondence: jane. mellor@bioch.ox.ac.uk

†These authors contributed equally to this work

‡These authors are joint second authors to this work

§These authors are joint third authors to this work

## Introduction

Genome-wide mapping of RNA transcripts in the budding yeast *Saccharomyces cerevisiae* has revealed an extensive array of coding and non-coding transcripts, giving rise to a genome that is heavily interleaved. Individual genes can possess multiple, overlapping transcripts, in the sense and the antisense orientations with respect to the pre-mRNA, as well as poly-cistronic transcripts, which span neighbouring genes (*Kapranov et al., 2007*; *Berretta et al., 2008*; *Pelechano et al., 2013*). Genome-wide mapping of nascent transcription, using techniques such as NET-seq, shows that transcription commonly extends into and over the intergenic regions of both convergent and tandemly arranged genes (*Churchman and Weissman, 2011*). For tandemly arranged genes, transcription into the promoter of the downstream gene would be expected to interfere with its expression by a variety of mechanisms, including modifying the local chromatin environment and interference by removal of transcription factors (*Martens et al., 2004*; *Martianov et al., 2007*; *Hainer et al., 2011*). Furthermore, extensive transcription antisense to (*Venters and Pugh, 2009*; *Murray et al., 2012*), and into the promoter of (*Perocchi et al., 2007*; *Xu et al., 2009*), the canonical coding transcript is also implicated in modulating gene expression by similar mechanisms (*Hongay et al., 2006*; *Camblong et al., 2007*; *Uhler et al., 2007*; *Houseley et al., 2008*; *Pinskaya et al., 2009*; *Xu et al., 2011*; *van Werven et al., 2012*; *Castelnuovo et al., 2013*). An interleaved genome with overlapping transcription units requires that polyadenylation and transcription termination signals in the sense and antisense orientations are

**eLife digest** A DNA double helix is made up of two DNA strands, which in turn are made of molecules that are each known by a single letter—A, T, C, or G. The sequence of these 'letters' in each DNA strand contains biological information.

Genes are sections of DNA that can be 'expressed' to produce proteins and RNA molecules. To express a gene, the DNA strands in the double helix must first be partially separated so that one of them can be used as a template to build an RNA molecule in a process called transcription. Either of the DNA strands in a helix can be used as an RNA template, but contain different genes and are read in opposite directions. One of the two strands is called the 'sense' strand, the other the 'antisense' strand.

The RNA molecule does not transcribe a whole DNA strand; instead, it transcribes a section of DNA, known as a transcription unit, which contains at least one gene. The end of a transcription unit is marked by certain signals that stop transcription. However, some transcription units in a DNA strand overlap, so there must be some way that the transcription machinery can sometimes ignore these stop signals.

The activity of some genes is linked to the activity of their immediate neighbours. Furthermore, some genes are expressed in different amounts in response to changes in environmental conditions. Researchers have previously suggested that there must be some form of switch that controls when these genes are expressed.

Nguyen et al. now engineer start and stop signals at a neighbouring pair of genes, called *HMS2* and *BAT2*, in yeast. When one gene is switched on, the other is switched off and which gene is active depends on the diet of the yeast cells.

On the antisense DNA strand opposite to *HMS2* is another gene, *SUT650*. Nguyen et al. show that when this gene is transcribed, the transcription of *HMS2* on the other DNA strand is blocked. This has the knock-on effect of turning on *BAT2*. Conversely, transcribing *HMS2* switches off *SUT650* and *BAT2* because the end of *HMS2* overlaps with the beginning of both *SUT650* and *BAT2*. Switching between different genes relies on loops that physically link the start and stop signals of the gene to be transcribed while ignoring the start and stop signals for neighbouring genes.

Proteins called transcription factors can bind to DNA and affect whether a gene is transcribed. Nguyen et al. found that a transcription factor that binds near the start of the *HMS2* gene helps to control which DNA strand is transcribed. When transcription factors do not bind to the start of *HMS2*, antisense transcription—and the expression of *SUT650*—occurs instead.

Overall, Nguyen et al. show that the transcription process itself makes up part of a switch that can control the expression of several genes on both the sense and antisense strands of a DNA double helix. This may also explain how many other, more complex, gene networks are activated in response to changes in the environment.

by-passed but it is not clear how this is achieved. In addition, questions are commonly raised about whether transcription of these overlapping transcription units is contemporaneous.

It is now clear that much transcription is organised into biologically relevant temporal windows within phenomena such as the metabolic cycle (*Tu and McKnight, 2006*). Indeed, periodic or cycling expression of genes can be detected using fluorescent reporters or dual-labelled RNA FISH in cultures of asynchronous cells (*Laxman et al., 2010*; *Silverman et al., 2010*), or in the absence of cell division (*Slavov et al., 2011*). This periodic expression is a result of synchronization of respiratory and glycolytic activities into robust oscillations in oxygen consumption, characterized by phase-specific transcript signatures involving over 3000 genes, known as the Yeast Metabolic Cycle (YMC) (*Klevecz et al., 2004*; *Tu et al., 2005, 2007*; *Soranzo et al., 2009*; *Slavov et al., 2011*; *Cai and Tu, 2012*). In the long-period YMC, a single cell alternates between periods of high (oxidative (OX) phase) or low oxygen consumption (reductive building (RB) and charging (RC) phases), the residence time in each phase being nutrient-dependent. For exponentially growing cells in batch culture, the majority of cells in the population will be in the OX phase of the YMC (*Slavov et al., 2011*). Transcript levels for genes that cycle in the YMC will change as the cell moves through these phases; at any time, some cells in an asynchronous population will contain a transcript and some will not. These shifts

through transcriptional states are robust but not invariable, as YMC-regulated genes switch on and off in response to cues prompted by both regulated and erratic changes in the intracellular and extracellular environment.

In addition to the partitioned gene expression patterns during the YMC, genomic spatial arrangements also contribute to regulatory relationships between genes, albeit on a smaller scale (*Cohen et al., 2000*; *Lee and Sonnhammer, 2003*; *Batada et al., 2007*). While genes displaying functional relatedness, such as the *GAL* genes, are regulated similarly by possessing the same *cis*-acting sequences (i.e. the UAS_{GAL}), co-expression of clustered genes in *S. cerevisiae* is largely independent of similarly controlled transcription, gene orientation, and/or shared regulatory sequences. However, members of adjacent gene pairs or clusters are more likely to belong to the same functional pathway than expected by chance (*Cohen et al., 2000*; *Lee and Sonnhammer, 2003*; *Batada et al., 2007*). The mechanism by which these clustered genes are regulated remains largely elusive.

Here we show that overlapping transcription, in both the sense and antisense orientations, constitutes an additional layer of regulation at clustered genes by managing state-switching in response to environmental change. We use a simple carbon source shift coupled with NET-seq to define genes whose transcription increases or decreases >threefold, after transfer from glucose (GLU)- to galactose (GAL)-containing media and show a remarkable enrichment for genes whose transcripts cycle during the YMC. The majority of these genes have no functional associations with transcription factors that might mediate repression or induction in GLU or GAL, but are organised in clusters and subject to overlapping sense and antisense transcription. We exemplify this mode of gene regulation, state-switching by transcriptional interference and insulation, at the *HMS2:BAT2* tandem gene cluster. By engineering promoters and terminators at *HMS2:BAT2*, we demonstrate the formation of alternative transcription units underlies state-switching. We suggest that overlapping transcription and the formation of alternative transcription units, associated with temporally segregated gene expression, will be a general feature of gene regulation.

## Results

### Genome-wide response to a change in carbon source reveals genes whose transcripts cycle in the YMC

The question we address in this work is whether transcriptional interference can explain the switching on and off of genes and thus altered gene expression in response to environmental change. We used a change in carbon source by shifting exponentially growing cells from glucose- (GLU) to galactose-containing media (GAL) for 3 h and analysed the native transcripts associated with elongating RNA polymerase II (NET-seq) (*Churchman and Weissman, 2011*) to identify genes whose transcription is altered during this environmental change (*Supplementary file 1A*). 10.45% (551) of ORF-Ts (open reading frame—transcripts) showed a >threefold increase and 9.99% (527) showed a >threefold decrease in transcription in GAL relative to GLU (*Supplementary file 1B*). By comparing the NET-seq output with a microarray of Poly(A)$^+$ RNA, we show that for the majority (≈88%) of genes, the change in transcript levels on the GLU to GAL shift reflects altered transcription rather than altered transcript stability (*Supplementary file 1A*). Gene ontology (GO) analysis revealed highly significant associations to growth, quiescence, and transcripts that cycle in the YMC, particularly during the oxidative (OX) and the reductive charging (RC) phases of the YMC (*Figure 1A*; *Supplementary file 1C,D*). The genes whose transcription decreases in GAL are enriched (467; 88.6% p < 1x10$^{-5}$) for OX phase genes that are mainly involved in ribosome biosynthesis and growth (*Figure 1B*). By contrast, for genes whose transcription increases in GAL, there is significant enrichment (392; 71.1%, p < 1 × 10$^{-5}$) for genes whose expression peaks in the reductive charging phase (RC) of the YMC (*Figure 1B*). These genes are associated with stress resistance, metabolism, and quiescence. 33.1% (891) of the 2691 annotated OX- and RC-regulated genes also show >threefold change on the GLU to GAL shift, supporting a shared regulatory mechanism between the YMC and the GLU to GAL shift (*Figure 1D*; *Supplementary file 1C*). In addition to ORF-Ts, many of the non-coding transcription events are regulated by the GLU to GAL shift. These include antisense transcription reads (to ORF-Ts), which increase in GAL compared to GLU (*Figure 1E*) and 25.7% of the 1772 annotated non-coding transcripts, the stable unannotated transcripts (SUTs), or cryptic unstable transcripts (CUTs), which show >threefold change in transcription on the GLU to GAL shift (*Supplementary file 1E*).

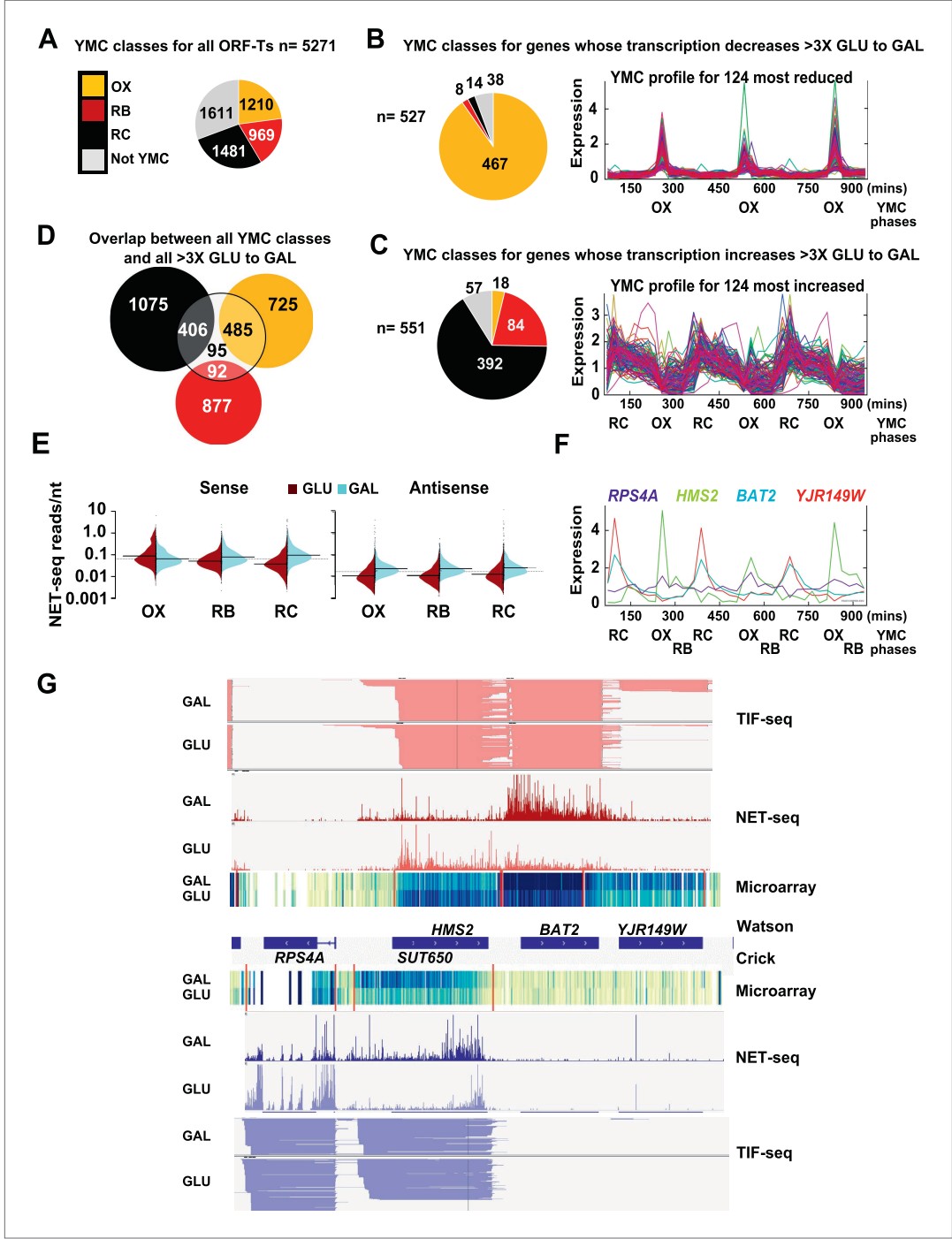

**Figure 1.** Reciprocal switching of expression by carbon source reveals links to the YMC. (**A**) The distribution of all ORF-Ts into different YMC phases. OX, oxidative phase; RB, reductive building phase; RC, reductive charging phase of the YMC. (**B**, **C**, **F**) YMC profiles (SCEPTRANS; http://moment.utmb.edu/cgi-bin/main_cc.cgi) (*Kudlicki et al., 2007*) showing cycling expression at 124 genes with the (**B**) most reduced and (**C**) most increased transcription (NET-seq) after a switch from glucose (GLU) to galactose (GAL) for 3 h and (**F**) at the *HMS2:BAT2* locus. (**D**) Overlap between all YMC classes and all ORF-Ts showing >threefold change in transcription on the GLU to GAL shift. (**E**) NET-seq reads on sense or antisense strands genome-wide in GLU (red) or GAL (blue) for ORF-Ts with peak expression in phases of the YMC indicated. (**G**) Strand-specific TIF-seq (*Pelechano et al., 2013*), microarray and NET-seq data at the *HMS2:BAT2* locus. Profiles from cells cultured in GLU or GAL on the Watson strand (top) or Crick strand (bottom) are shown. The TIF-seq data indicate each different transcript isoform, microarray data

*Figure 1. Continued on next page*

*Figure 1. Continued*

indicate levels of steady-state poly(A)$^+$ RNA (blue—darker colour for more RNA) and the NET-seq data (normalized, unique, and clipped to 3'OH) display transcript reads (scale 0–30) associated with elongating RNAPII. Screen shots are displayed using IGV (*Robinson et al., 2011*; *Thorvaldsdottir et al., 2013*).

The following figure supplements are available for figure 1:

**Figure supplement 1**. Exemplary tandem gene clusters and their regulation by carbon source and the YMC.

**Figure supplement 2**. Exemplary tandem gene clusters and their regulation by carbon source and the YMC.

**Figure supplement 3**. Exemplary tandem gene clusters and their regulation by carbon source and the YMC.

**Figure supplement 4**. Exemplary tandem gene clusters and their regulation by carbon source and the YMC.

## Global analysis reveals distinct organisation and regulation

We used a computer simulation to obtain the expected number of occurrences of genome-wide features, by randomly shuffling genes, their orientation, YMC status, and expression levels (data from *Supplementary file 1A*, codes in *Source code 1*). Selected data from this analysis are shown in *Supplementary file 1F*. In summary, pairs of ORF-Ts in the convergent orientation occur more often than expected, while tandemly arranged ORF-Ts occur less often than expected. YMC genes are clustered, with 503 (16.9%) YMC genes flanked by non-cycling genes, 1431 (48.4%) with a single adjacent YMC partner either upstream or downstream and 1023 (34.5%) with a neighbouring cycling gene both upstream and downstream. Compared to more isolated genes, ORF-T clusters are more likely to show increased transcription in GLU or GAL than expected, suggesting that clustering is associated with some aspect of their regulation. In addition, ORF-Ts flanked by annotated SUTs or CUTs (in any orientation) occur more often than expected and show a twofold to fourfold higher median expression upon change from GLU to GAL than the average median expression obtained through the simulations. This is consistent with CUTs and SUTs playing a role in modulating ORF-T transcription, particularly on environmental change (*Xu et al., 2011*). Convergent overlapping arrangements occur three times more frequently than expected, whether the feature is a non-coding transcript (NC) such as a SUT or a CUT, or an ORF-T, with a median overlap of 92 bp for convergent ORF-Ts and 462 bp for ORF-Ts with non-coding transcripts. Thus clustering with overlapping transcription and transcriptional regulation are common features of the yeast genome.

As the genes whose transcription changed during the GLU/GAL shift are enriched for OX and RC YMC genes, we examined whether the nature of the flanking gene influences transcription, focusing on the tandem, and divergent combinations of these genes (*Supplementary file 1F*). OX genes, regardless of the nature (OX or RC) or orientation (divergent or tandem) of the upstream feature showed higher than expected transcriptional rates in GLU, but not in GAL, suggesting they are activated in GLU but not repressed in GAL. Similarly, RC genes are more likely to show higher than expected transcriptional rates in GAL. The behaviour of RC genes in GLU, however, does appear to be dependent on the orientation of the upstream OX gene. With a divergent upstream OX gene, the transcription of the downstream RC pair is not enriched or depleted in GLU. However, if the OX gene is in tandem, the RC gene is repressed in GLU. This raises the possibility that at the OX.RC tandem pair, transcription of the OX gene is mediating transcriptional interference and repressing the transcription of the RC gene in GLU. We note that the downstream gene in a tandem RC.RC pair also shows significant repression in glucose. To investigate further, we examined three different data sets: (i) three groups of tandemly arranged genes; OX.RC, RC.OX, and non-cycling (*Supplementary file 1G*); (ii) three sets of 20 consecutive ORF-Ts from *Supplementary file 1A* showing a sevenfold increase, no change or a sevenfold decrease on the GLU/GAL shift (*Supplementary file 1H*); and (iii) GLU/GAL-regulated genes with an annotated antisense CUT or SUT that is also GLU/GAL-regulated (*Supplementary file 1I*). From this analysis, we picked tandem gene clusters where the transcription of the target gene is regulated by GAL/GAL shift or not, and examined how neighbouring genes behave (*Figure 1—figure supplements 1–4*; *Supplementary file 1H*; *Supplementary file 1J*). The tandem partner of a GLU/GAL-regulated gene is often also regulated by the GLU/GAL shift and the YMC, while non-regulated genes are less likely to be in tandem and if they

are, are surrounded by genes that are also not regulated by the GLU/GAL shift or the YMC. Common features of the regulated tandem clusters include reciprocal transcription in GLU and GAL, reciprocally cycling transcripts in the YMC, reciprocal AS transcription to OX genes, di-cistronic transcripts, and/or antisense transcripts spanning promoters that could mediate temporal transcriptional interference and thus facilitate cycling transcription. We chose the OX.RC pair *HMS2:BAT2* for further study, as all the above characteristics are present and because the relatively abundant *HMS2* antisense transcript, *SUT650*, can be detected experimentally in both GLU and GAL (*Figure 1F,G*).

## Characterisation of transcripts and transcription at the *HMS2:BAT2* locus

We examined the relationship between *HMS2* and *BAT2* (transcripts A and D respectively). *BAT2* transcripts peak in the RC phase of the YMC and in GAL, reciprocal to *HMS2* sense transcripts, which peak in the OX phase of the YMC and in GLU (*Figure 1F,G* and *Figure 2—figure supplement 1A–C*). Interestingly, there is also a reciprocal relationship between *HMS2* and its antisense *SUT650* (transcripts A and B respectively). Dual-labelled RNA FISH revealed a degree of temporal separation in the production of the *HMS2* and *SUT650* transcripts. In multiple analyses, no examples of FISH signals for both *HMS2* and *SUT650* at the site of transcription in the nucleus are observed in either GLU or GAL. Cells with nascent transcripts (arrows, *Figure 2A*) often have the opposite transcript type or both *SUT650* and *HMS2* transcripts in the cytoplasm, suggesting they are changing state. 35.5 ± 7.9% of the cells contained both transcripts in GLU, 22.8 ± 10.5% of all cells lacked both the *SUT650* and *HMS2* sense transcripts. The remaining cells contained either *HMS2* or *SUT650* transcripts but not both (*Figure 2A*). For cells containing a single type of transcript, more cells contained antisense transcripts in GAL than GLU while more cells contained sense transcripts in GLU than GAL (*Figure 2B*). Thus, the increase in levels of *SUT650* transcripts in GAL reflects an increase in the number of cells expressing *SUT650* and a concomitant decrease in the number of cells expressing *HMS2*, rather than a change in the levels of transcripts within a fixed number of cells in the population. We conclude that the expression of *HMS2* and *SUT650* is temporally separated, similar to the *PHO84* sense and antisense transcripts (*Castelnuovo et al., 2013*). This suggests potential co-regulation between *BAT2* and *SUT650* and reciprocal regulation of *HMS2* and *BAT2*.

A number of different methods revealed heterogeneity at both the 5′ and 3′ ends of the *HMS2* and *SUT650* transcripts and longer RNA species ($C_1$ and $C_2$) on the *HMS2* sense strand extending through the coding region of *BAT2* and beyond, consistent with low-level read-through transcripts initiating at the *HMS2* promoter (*Figure 2C–E* and *Figure 2—figure supplement 1D,E*). These read-through transcripts increase upon loss of nuclear exosome (*rrp6Δ*) (*Figure 2D*). Thus *HMS2* is at the head of a contiguous gene cluster including the downstream genes *BAT2* and *YJR149W* (*Figure 2E*). The 3′ UTR of the *HMS2* transcript is of variable length. Polyadenylation (pA) occurs either 85 nt (proximal) or 426 nt (distal) downstream from the *HMS2* stop codon. The distal pA site is 45 nt upstream of the *BAT2* initiation codon, such that this longer transcription event is likely to interfere with initiation of *BAT2* transcription. Additional transcriptional interference might be mediated through the production of the longer *HMS2-BAT2* di-cistronic RNA species.

In summary, analysis of transcripts around *HMS2* and *BAT2* supports (i) an overlapping, reciprocally related and temporally segregated, sense–antisense pair (SAP) at *HMS2*, (ii) reciprocally expressed sense transcripts at *HMS2* and *BAT2*, and (iii) low-level read-through di-cistronic transcripts (*HMS2:BAT2* and *SUT650:RPS4A*). We ask if overlapping transcription in the sense and antisense orientations at *HMS2* contributes to the reciprocal regulation of *HMS2* and *BAT2*, by dissociating one transcription unit from the other.

### *HMS2* transcription interferes with *BAT2* transcription

Insertion of the *ADH1* transcription terminator (*ADH1*t) into the middle to *HMS2*, to separate *HMS2* transcription from *BAT2*, allows us to test directly whether the *HMS2* sense transcription over the *HMS2:BAT2* intergenic region represses *BAT2* transcription (*Figure 3A*). We confirmed that *ADH1*t efficiently terminates the *HMS2* transcript (*Figure 3B,C*) and that there are no detectable transcripts over the *HMS2* 3′ region (*Figure 3D*). The strain with *ADH1*t inserted showed a ≈twofold increase in *BAT2* transcript levels compared to WT in both GLU and GAL (*Figure 3B,C*) supporting a role for *HMS2* sense transcription in repressing the *BAT2* promoter. We conclude that *HMS2* sense transcription interferes with *BAT2* transcription.

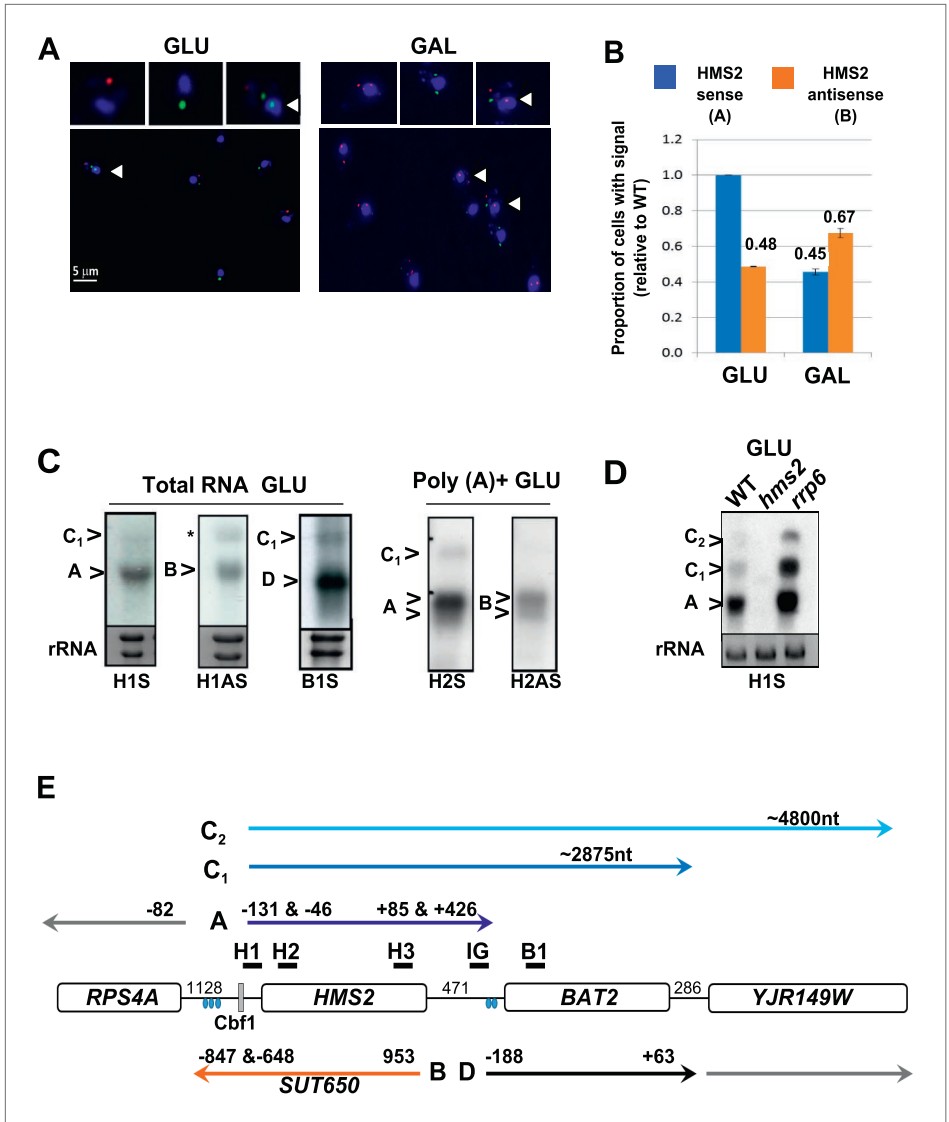

**Figure 2**. Characterization of transcripts around the *HMS2:BAT2* locus (**A**). Visualizing *HMS2* transcripts using RNA fluorescence in situ hybridization (RNA FISH) in single cells using a combination of four, 50 nt DNA probes labelled with four fluorophores, either Cy5 (sense) or Cy3 (antisense), hybridised to paraformaldehyde-fixed yeast cells. DAPI (blue) marks the nucleus. Smaller boxes are zoomed images of selected cells in the field of view. Cells with nascent nuclear transcription event are marked with arrows. The images presented here are part of larger data sets. The graph (**B**) represents the proportion of sense- and antisense-expressing cells between growth conditions from different experiments. A total of ≈250 cells were assessed in each growth condition. The mean (MAX) signal from an *hms2Δ* strain prepared at the same time was used to threshold the signal intensity. Error bars are SEM, n = 2, *Figure 2—source data 1A*. (**C**) Northern blots of total and poly(A)+ selected RNA from WT cells probed with *HMS2* (**H**) and *BAT2* (**B**) sense (S)- and *HMS2* antisense (AS)-specific probes (positions indicated in *Figure 2E*). Note the *HMS2* antisense probe shows cross-hybridization with the 25S rRNA and is marked * in this and subsequent figures. Ethidium bromide-stained rRNA is used as a loading control. (**D**) Northern blot of total RNA showing *HMS2* sense transcripts in strains indicated. (**E**) Map of transcripts (**A** to **D**) around the *HMS2* locus showing approximate length, initiation sites, and termination sites relative to the ATG for each gene (see also *Figure 2—figure supplement 1*). The Cbf1 transcription factor binding site is shown (grey boxes). Different transcripts are coloured: blue for *HMS2*, orange for *SUT650*, black for *BAT2*, and grey for *RPS4A* and *YJR149W*. Transcripts are shown on the top or bottom of the schematic to reflect linked expression. Thick black lines indicate probe position for Northern blots, with the name of the probe above.

*Figure 2. Continued on next page*

*Figure 2. Continued*

The following source data and figure supplement is available for figure 2:

**Source data 1**.
**Figure supplement 1**. Characterization of transcripts around the *HMS2:BAT2* locus.

## *SUT650* transcription limits the number of cells expressing *HMS2*

Insertion of the *ADH1*t into the *HMS2* coding region resulted in a ≈fourfold increase in levels of a truncated *HMS2* sense transcript ($A^A$) (*Figure 3A,B,D*) and blocks *SUT650* antisense transcription, resulting in the truncated antisense transcript ($B^A$) (*Figure 3A,D*). We confirm there are no antisense transcripts over the 5′ region of *HMS2* (*Figure 3B,E*). In the absence of *SUT650*, expression of the *HMS2:ADH1t* transcript remains sensitive to the change in carbon source, showing a similar decrease to full-length *HMS2* 1 h after transfer from GLU to GAL (*Figure 3B,C*). This suggests that the carbon source responsive signal is received at the *HMS2* promoter and that reciprocal switching of *HMS2* and *SUT650* is a function of sense transcription. However, as the levels of *HMS2* sense transcripts are higher without the antisense transcript, we asked if antisense transcription plays a role in limiting the number of cells containing sense transcript using RNA FISH (*Figure 3F*). When grown in glucose, optimal conditions for *HMS2* sense transcription, we observed an increase in the number of *HMS2:ADH1t* cells containing sense transcript compared to WT (*Figure 3G*), consistent with a role for antisense transcription in reducing the number of sense transcription initiation events, leading to a decrease in the number of cells in the population that contain the sense transcript. We conclude that *SUT650* antisense transcription does not influence the environmental responsiveness of *HMS2* transcription but acts to limit the number of cells in the population that respond.

## *SUT650* antisense transcription insulates *BAT2* from interference by *HMS2*

Next, we asked whether transcription of *SUT650* antisense might prevent *HMS2* sense transcription from interfering with *BAT2*, thus insulating *BAT2*. To ablate *SUT650* whilst ensuring that the regulatory elements in the flanking sequences between *HMS2* and *BAT2* remained intact, we removed the transcription start sites (TSS) for *SUT650* located between +953 and +1038 relative to the *HMS2* ATG, near the end of the *HMS2* coding region. We chose to do this by substituting the complete coding region of *HMS2* with that of *URA3*, to ensure a natural ORF between the *HMS2* promoter and terminator, rather than mutating the regions containing multiple TSSs for *SUT650* within the ORF (*Figure 4A*). We note that insertion of the *URA3* ORF could affect transcript stability, although the nature of the promoter may be the primary determinant of transcript stability (*Bregman et al., 2011*; *Trcek et al., 2011*), so we focused on transcript size and relative abundance. We observed two clear effects. First, loss of the antisense transcript (B), which could be explained either by introduction of a unidirectional terminator in the *URA3* ORF coding region or loss of TSSs for *SUT650* (*Figure 4B–D* and *Figure 4— figure supplement 1*). Second, an increase in the read-through transcripts ($C_1^U$) relative to the *HMS2:URA3* sense transcript ($A^U$) (*Figure 4B–D*). We confirmed these were read-through transcripts by inserting a terminator (T) at the 3′ region of the *HMS2:URA3* hybrid gene (*Figure 4A*) and observing loss of the read-through transcripts and increased levels of a shorter transcript ($A^U$T) (*Figure 4E,F*). We note that transcripts initiated at the *HMS2* promoter remain sensitive to a change in carbon source in the absence of antisense transcription (*Figure 4D*), as we observed using *HMS2:ADH1t* (see *Figure 3*) and consistent with a regulatory input at the promoter. The increase in the read-through transcript ($C_1^U$) relative to the *HMS2:URA3* sense transcript ($A^U$) in the absence of antisense transcription suggests a role for antisense transcription in attenuating sense transcription and preventing interference at the *BAT2* promoter. Consistent with this, a twofold decrease in *BAT2* sense transcript is observed in the antisense-less *HMS2:URA3* strain relative to WT in GLU, concomitant with a threefold increase in $A^U$ and $C_1^U$ (*Figure 4B,C*). Taken together, the effect of blocking (*Figure 3*) or increasing (*Figure 4*) *HMS2* transcription on levels of *BAT2* transcript supports transcriptional interference by *HMS2* at the *BAT2* promoter and a reciprocal relationship between *HMS2* and *BAT2*. We sought evidence for transcriptional interference resulting from increased levels of $A^U$ and $C_{1,2}^U$. Chromatin immunoprecipitation (ChIP) analysis at the *HMS2:BAT2* intergenic region reveals reduced initiation-related modifications

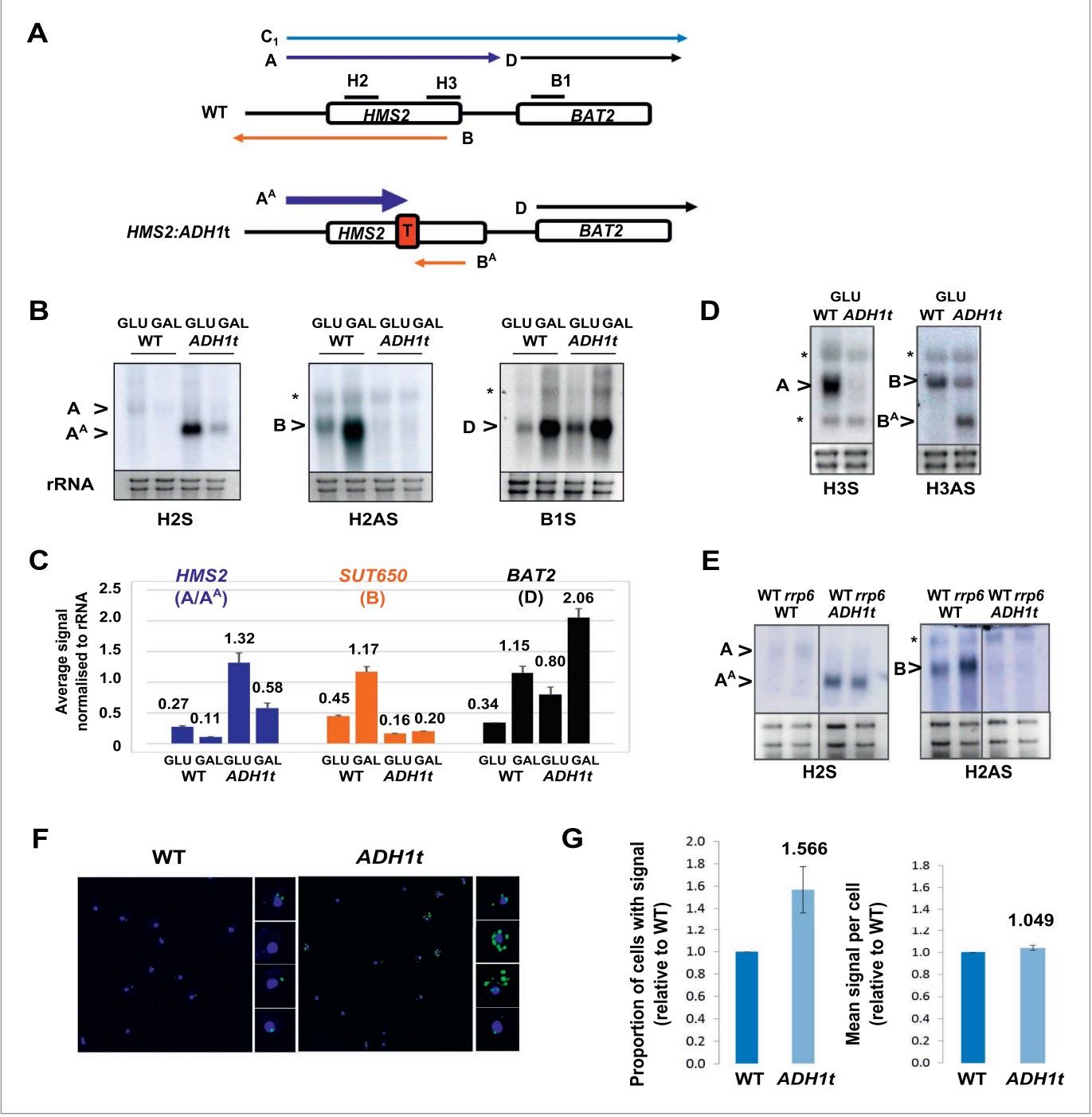

**Figure 3**. *HMS2* mediates transcriptional interference of *BAT2*. (**A**) Schematic showing constructs and transcripts at the WT *HMS2:BAT2* locus and after insertion of the *ADH1* terminator (T). (**B**) Exemplarily autoradiographs of Northern blots of total RNA prepared from the constructs in (**A**) cultured in glucose or after 60 min in galactose probed for the *HMS2* sense, the *SUT650* antisense, and the *BAT2* sense transcripts. (**C**) Quantitation of autoradiographs showing average signal normalized to rRNA for the transcripts indicated. n = 2, errors are SEM, *Figure 3—source data 1A*. (**D**, **E**) Exemplarily autoradiographs of Northern blots of total RNA prepared from the strains indicated containing the constructs in (**A**) cultured in glucose probed for the regions indicated. (**F**) Visualizing *HMS2* sense transcripts using fluorescence in situ hybridization (FISH) in single cells using a combination of four, 50 nt DNA probes labelled with four Cy5 fluorophores (green, sense), hybridised to paraformaldehyde-fixed yeast cells. The nucleus is shown in blue (DAPI). Smaller boxes are zoomed images of select cells in the field of view. The images presented here are part of a larger data set. (**G**) The graphs represent the proportion of sense-expressing cells in the WT compared to *HMS2:ADH1t* and the mean signal per cell. A total of ≈500 cells were assessed for each strain. Error bars are SEM; n = 4, *Figure 3—source data 1A,C*).

*Figure 3. Continued on next page*

*Figure 3. Continued*

The following source data are available for figure 3:

**Source data 1**.

(H3K4me3 and H3K56ac) and increased elongation-related modifications (H3K79me3 and H3K36me3) in the *HMS2:URA3* hybrid gene, consistent with transcriptional interference leading to repression of *BAT2* transcription initiation (*Figure 4G*). We conclude that *SUT650* antisense transcription (i) changes the proportion of cells expressing *HMS2* and (ii) controls the amount of overlapping and read-through transcription from *HMS2* and thus levels of *BAT2* transcript. We envisage that at any one time in a single cell, formation of the *HMS2* and *SUT650* transcription units (TUs) is mutually exclusive, but the formation of the *SUT650* TU and the *BAT2* TU is not. Thus formation of the *SUT650* transcription unit insulates *BAT2* from interference by *HMS2* transcription, as the *HMS2* TU cannot form when *SUT650* is expressed.

### *HMS2* transcription represses the *SUT650* antisense transcript and *BAT2*

To address whether *HMS2* transcription influences *SUT650* antisense transcription, we replaced the *HMS2* promoter with the inducible *GAL1* promoter (p*GAL1*) (*Figure 5A*). When cultured in glucose, the *GAL1* promoter is repressed but becomes induced about 2 h after transfer to galactose (*Figure 5B*). *SUT650* is normally induced during the GLU-GAL shift (see *Figure 1*). However in the p*GAL1:HMS2* strain, an antisense transcript (B$^G$) is evident in GLU and remains until the sense transcript is induced by GAL (*Figure 5C,D*). This suggests that sense transcription represses the antisense transcript. We used the inducible p*GAL1:HMS2* system in conjunction with the Anchor Away technique (*Haruki et al., 2008*) to deplete the nucleus of the Gal4p activator during growth in GAL. Removal of Gal4p by treatment with rapamycin (Rap) for 1 h, after activation of p*GAL1:HMS2* for 3 h with GAL, abolishes the *HMS2* transcript (A$^G$) and leads to restoration of the antisense transcript (B$^G$) (*Figure 5E,F*). This confirms that the reduction in the antisense transcript is a consequence of sense transcription, rather than the change in condition (GLU to GAL). To determine whether antisense transcription had any effect on p*GAL1:HMS2* sense transcript induction rates, the antisense--less *HMS2:URA3* construct was placed under the control of the *GAL1* promoter (*Figure 5A,B*). The sense transcript profile in poly(A)$^+$-enriched RNA was examined using a probe for the *HMS2:BAT2* intergenic region (probe IG), a region transcribed in both constructs. No changes in induction rate were apparent, with the sense transcript only detectable after 2 h in both strains, signifying that antisense transcription does not impair sense activation kinetics (*Figure 5B*). Rather, this finding is compatible with our observation that without antisense, more cells would express the sense transcript (see *Figure 3F,G*). Activation of the p*GAL1:HMS2* sense transcript results in up to 3.5-fold repression of *BAT2* (*Figure 5F*), consistent with p*GAL1:HMS2* sense polyadenylation at 426 nt, just upstream of the *BAT2* ATG, interfering with *BAT2* transcription initiation in galactose (see *Figure 2—figure supplement 1*). We conclude that formation of the *HMS2* sense transcription unit represses the formation of the *SUT650* and *BAT2* transcription units, regardless of conditions.

### Antisense transcript isoforms are condition-specific and inherent to *HMS2*

There are two species of antisense transcript during induction of p*GAL1:HMS2*; a long species (B$^{G1}$), dominant until 30 min after the shift to GAL, and a shorter species (B$^{G2}$), also detectable after 30 min of growth in GAL and which becomes the dominant antisense transcript form after 1 h (*Figure 5C*). 3'end RACE revealed alternative polyadenylation sites as one cause for the observed size difference, with the long form terminating upstream of the Gal4p binding sites at approx. 500 nt upstream of the ATG in the *GAL1* promoter and the shorter form terminating midway through the *GAL1* promoter, 195 nt upstream of the *GAL1* ATG, and downstream of the Gal4p binding sites (*Figure 5—figure supplement 1*). Upon glucose repression of induced p*GAL1:HMS2*, levels of the long form of antisense transcript were recovered (*Figure 5G*). Antisense transcript shortening after transfer to GAL is an inherent property of the natural *SUT650* antisense transcript (*Figure 5H*). To ask if this was a function of transcription factor binding to the promoter, we used the p*GAL1:HMS2* construct and showed

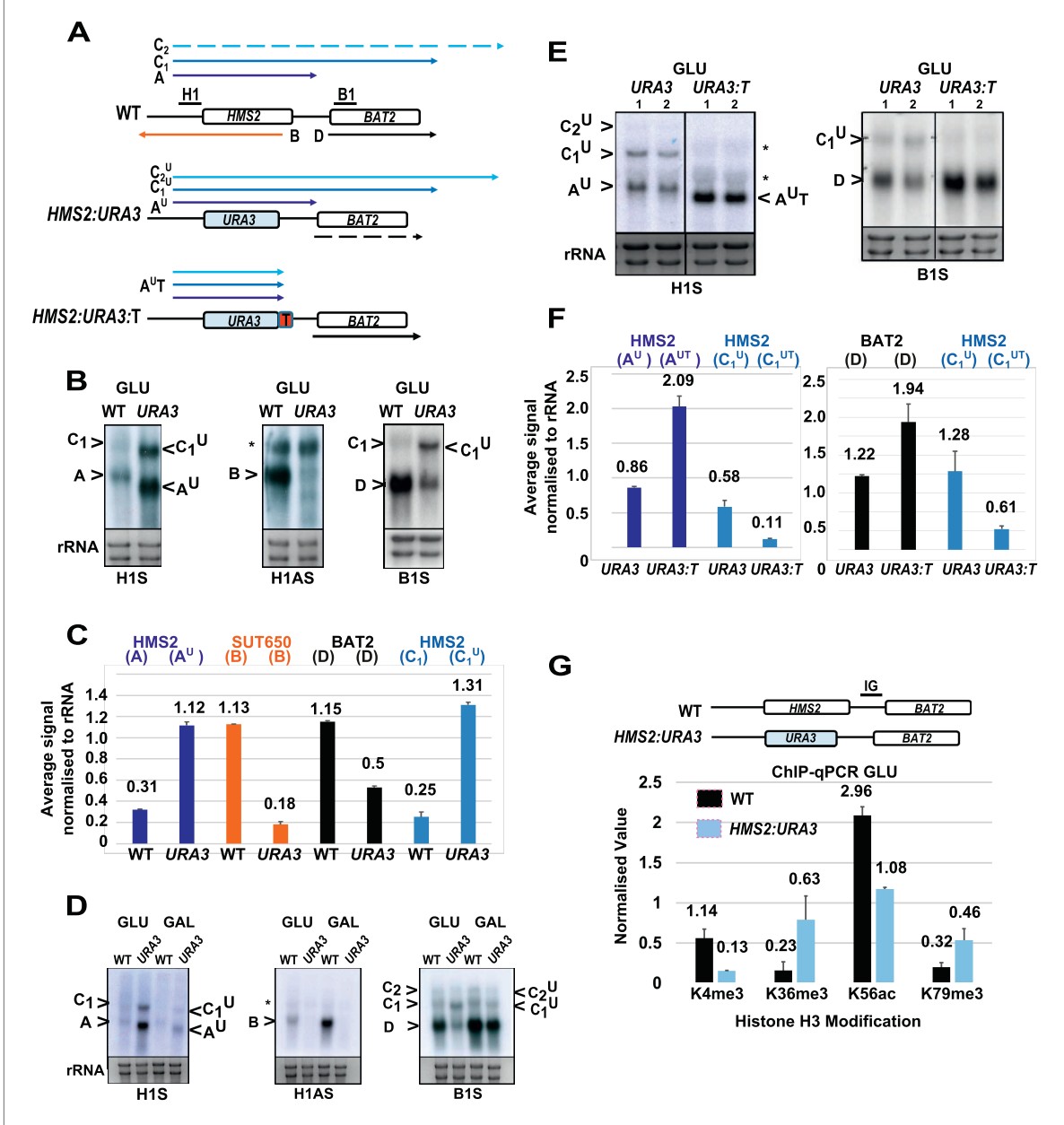

**Figure 4**. *SUT650* antisense transcription insulates *BAT2* from interference by *HMS2*. (**A**) Schematic showing transcripts after replacement of the *HMS2* coding region (top) with the *URA3* coding region (middle) or *URA3* plus a transcription terminator (T) (bottom). Transcripts resulting from the *URA3* insertions are identified with a superscript U or UT. (**B**, **D**, **E**) Northern blots of total RNA in strains with the *HMS2* coding region replaced with the *URA3* coding region in glucose (**B**), after 1 h in galactose (**D**) or in glucose with a terminator (T) inserted after *URA3* (**E**). In (**E**), samples were run on the same gel but intervening tracks removed. (**C**) Quantitation of transcripts for WT and *HMS2:URA3* in glucose, n = 2, errors are SEM, *Figure 4—source data 1A*. (**F**) Quantitation of transcripts in (**E**) for *HMS2:URA3* and *HMS:URA3:T* in glucose, n = 2, errors are SEM, *Figure 4—source data 1B*. (**G**) Chromatin immunoprecipitation (ChIP-qPCR) at the *HMS2:BAT2* intergenic region (IG) in strains indicated using antibodies with the specificities indicated. Signals were normalized to Histone H3 and then the signal in the coding region of *TUB2*. Error bars are SEM for n = 2; *Figure 4—source data 2*.

The following source data and figure supplement is available for figure 4:

**Source data 1**.

**Source data 2**.

**Figure supplement 1**. No *SUT650* antisense transcription in either GLU or GAL in the *HMS2:URA3* strain.

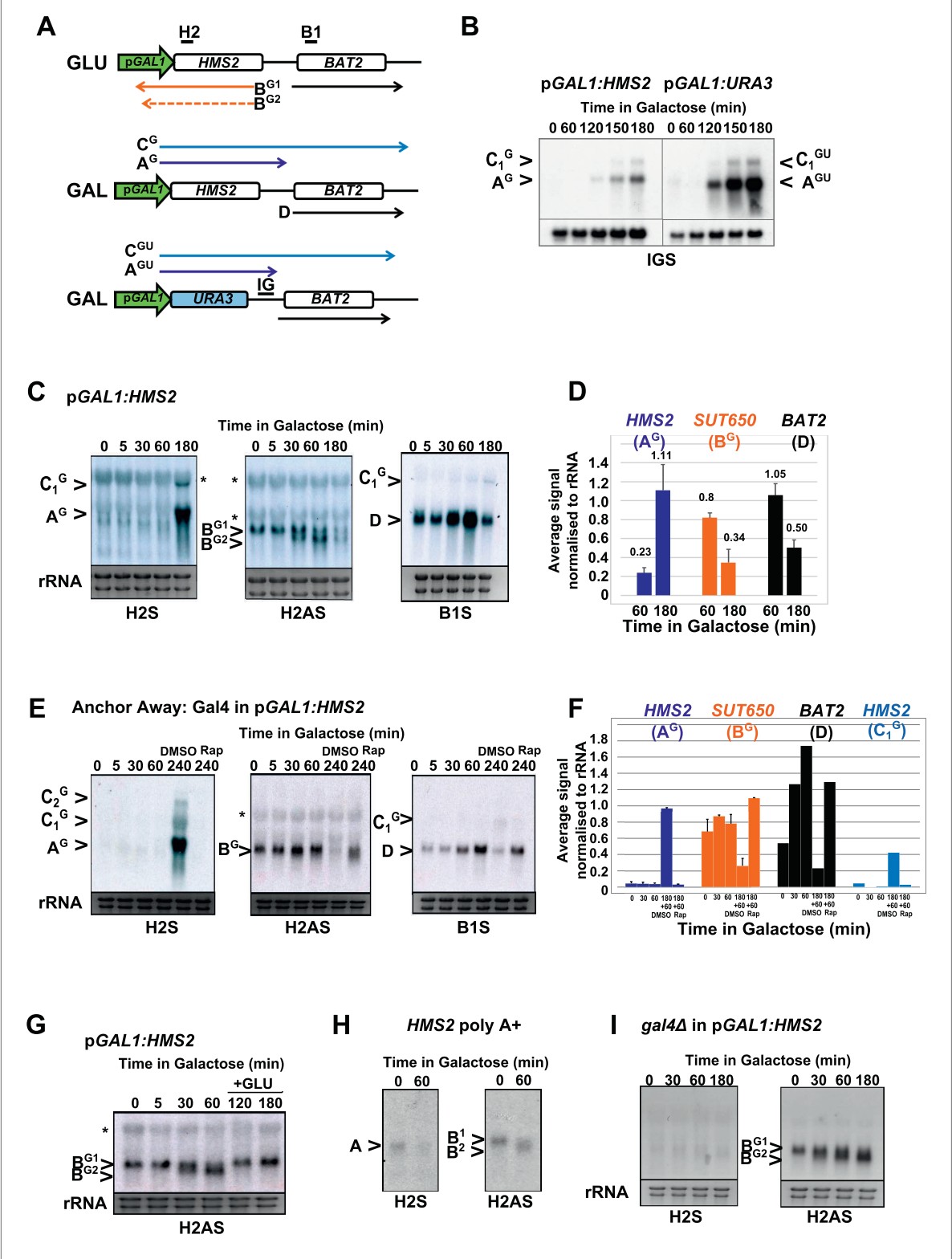

**Figure 5**. *HMS2* sense transcription represses the *HMS2* antisense transcript and *BAT2*. (**A**) Schematic showing transcripts when expression of *HMS2* is regulated by the *GAL1* promoter (p*GAL1*) in glucose (antisense-dominant state) or galactose (sense-dominant state) or when the *HMS2* coding region is substituted with the *URA3* coding region. (**B**, **C**, **E**, **G–I**) Northern blots showing total RNA (**C**, **E**, **G**, **I**) or poly(A)$^+$ RNA (**B**, **H**) from cells expressing p*GAL1:HMS2* cultured in glucose (GLU) or induced for the times indicated in galactose (GAL) in the genetic backgrounds indicated. (**D**, **F**) Quantitation

*Figure 5. Continued on next page*

*Figure 5. Continued*

of the experiments exemplified in (**C**) and (**E**) respectively, n = 2, error are SEM, *Figure 5—source data 1A,B,C*. (**E**, **F**) Anchor Away of Gal4 is achieved after incubation with rapamycin in DMSO for 1 h, added 3 h after induction with galactose (240 Rap). The control culture (240 DMSO) was treated with DMSO. See also *Figure 5—figure supplement 1*.

The following source data and figure supplement is available for figure 5:

**Source data 1**.

**Figure supplement 1**. Mapping the 3′ ends of the p*GAL.HMS2* antisense transcript isoforms.

that shortening of the antisense transcript is independent of Gal4 binding at p*GAL1* and remains for 3 h in GAL in the absence of sense transcription (*Figure 5I*). This rules out a potential roadblock by Gal4 as an explanation for the use of different antisense polyadenylation/termination sites in p*GAL1*. We conclude that *SUT650* antisense transcription is normally repressed by *HMS2* transcription but remains stably expressed in GLU or GAL, varying in the use of polyadenylation sites, when sense transcription is inhibited.

## Transcription factors contribute to sense/antisense switching at *HMS2:SUT650*

To see if loss of transcription factors associated with the native *HMS2* gene, Cbf1 (*MacIsaac et al., 2006*), and Fkh1 (*Venters et al., 2011*), influence the balance of sense and antisense transcription at *HMS2*, we used *cbf1Δ*, *fkh1Δ*, and *fkh2Δ* strains, regulators of nutrient availability, oxidative growth, and the cell cycle (*Mitchell and Magasanik, 1984*; *Mellor et al., 1990*; *Kent et al., 1994*; *Zhu et al., 2000*; *Morillon et al., 2003*; *Tsankov et al., 2011*). While under standard growth conditions in GLU these mutant strains showed normal expression of *HMS2* and *SUT650*, growth in nutrient-limited GLU-based medium depleted for tryptophan resulted in loss of the *HMS2* transcript and accumulation of the short isoform of the *SUT650* antisense transcript (B$_2$) in the *cbf1Δ* and *fkh1Δ* strains (*Figure 6A*). In contrast, the *fkh2Δ* strain showed reduced levels of *SUT650* and increased *HMS2* under these growth conditions. This supports transcription factors controlling the balance of sense, and thus antisense transcription, at this locus during the cell cycle (Fkh factors) (*Granovskaia et al., 2010*) and during oxidative growth in metabolic cycles (Cbf1) (*Tsankov et al., 2011*). We ablated a number of non-essential transcription factors, including Gln3, with putative binding sites at the 3′ region of *HMS2* (*MacIsaac et al., 2006*) but observed no significant change in levels of *SUT650*, *HMS2*, or *BAT2* transcripts with the exception of a small increase in the *HMS2:BAT2* read-through transcript (C) in a *gln3Δ* strain in minimal medium (*Figure 6—figure supplement 1*). We propose that transcription factors signal to the *HMS2* promoter (Input) to mediate state-changing (antisense-dominant to sense-dominant) in response to environmental conditions (*Figure 6B*). As we could find no direct evidence for regulation of *SUT650* either by TFs (*Figure 6* and *Figure 6—figure supplement 1*) or by the GLU to GAL shift (see *Figure 5*) but only by transcriptional interference resulting from activation of *HMS2* transcription, we examined OX and RC genes genome-wide to look for common themes in their regulation. We asked what factors are significantly enriched (p < 0.01) at promoters of RC and OX genes, using a large data set of 202 factors (*Venters et al., 2011*). We observed marked differences (*Supplementary file 1K*). Of particular interest are the predominance of chromatin modulators and components at RC gene promoters, suggesting that RC promoters are particularly sensitive to chromatin-mediated repression (*Tirosh and Barkai, 2008*). These features, coupled with the distinct behaviour of OX and RC genes in GLU and GAL, particularly OX.RC genes in tandem (*Supplementary file 1F*), lead us to propose chromatin-mediated repression of RC genes by transcriptional interference. In addition, antisense transcription into OX promoters, for example by *SUT650* at *HMS2*, may also limit OX gene transcription.

## TFIIB (Sua7) is required to limit the sense-dominant state at *HMS2*

We assessed a role for a number of essential transcription factors using the Anchor Away technique (*Haruki et al., 2008*) but only upon nuclear depletion of TFIIB (Sua7) did we observe an increase in the *HMS2:BAT2* read-through transcript (C) (*Figure 6C*). However, this occurred together with reduced levels of *HMS2* (A), *BAT2* (D), and *SUT650* (B) transcripts, as expected upon depletion of an essential transcription factor. To explore this further, we used a well-characterised allele of *SUA7* (*sua7-1*) (E62K)

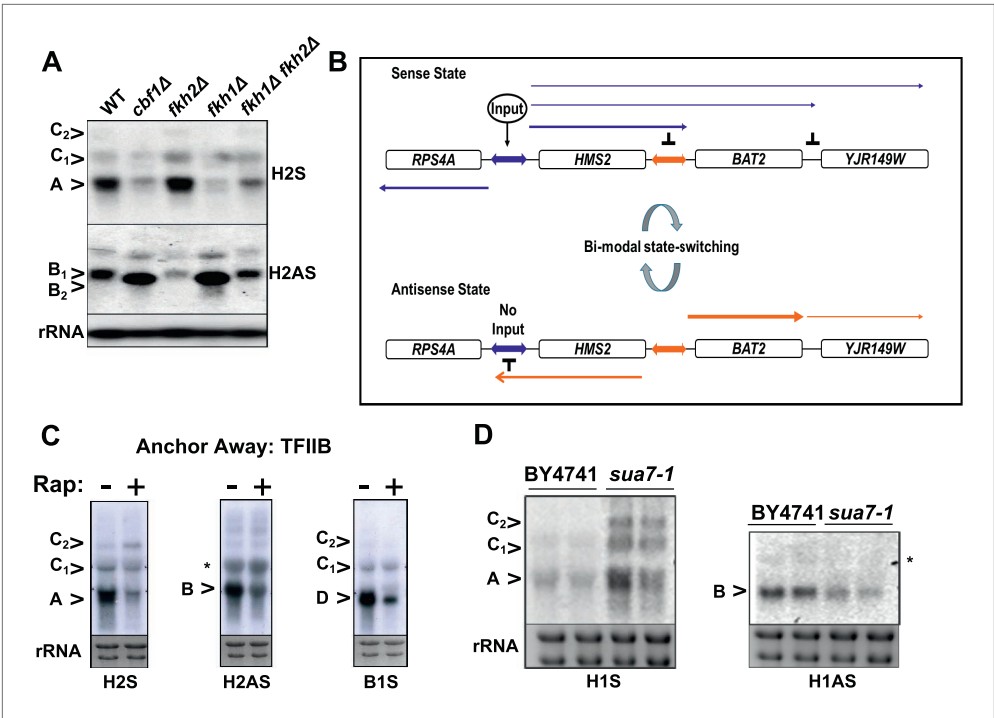

**Figure 6**. General and specific transcription factors control state switching at *HMS2*. (**A**, **C**, **D**) Northern blots showing sense or antisense-dominant state at *HMS2* in strains lacking general or specific transcription factors. (**A**) Strains were cultured in YPD depleted for tryptophan. (**B**) Model for state-switching between a sense-dominant state and an antisense-dominant state by transcription factors (Input) at the divergent promoters between *RPS4A* and *HMS2* (blue double arrow) and *HMS2* and *BAT2* (orange double arrow). During growth in glucose (high input), cells cycle between the sense-dominant state and the antisense-dominant state with the majority of cells existing in the sense-dominant state. During growth in galactose (low input), cells cycle between the sense-dominant state and the antisense-dominant state with the majority of cells existing in the antisense-dominant state. (**C**) Anchor Away of TFIIB (Sua7) is achieved after incubation with rapamycin (Rap) (+) in DMSO for 1 h or DMSO alone (−). (**D**) Biological replicates for the WT strain BY4741 and the isogenic strain expressing the *sua7-1* allele are shown.

The following figure supplement is available for figure 6:

**Figure supplement 1**. The effect of deletion or ablation of transcription factors with putative binding sites at the *HMS2:BAT2* intergenic region on *HMS2:BAT2* transcripts.

---

(*Pinto et al., 1994*), associated specifically with the 3′ region of genes, polyadenylation/transcription termination and higher order structures (gene loops) in chromatin (*Medler et al., 2011*). Remarkably, we observed an increase in *HMS2* transcript (A) and sense read-through transcripts (C), coupled with a decrease in the *SUT650* antisense transcript (B) (*Figure 6D*). Thus, TFIIB may have a specific role in determining the antisense transcription unit by establishing directionality of transcription over the region (*Tan-Wong et al., 2012*). For instance, by determining an antisense transcription unit, TFIIB aids in sense transcription termination, preventing 'read-through' over the intergenic region between *HMS2* and *BAT2*. The capacity to read through transcription regulatory regions such as promoters and terminators is an essential component of a model for gene regulation by transcriptional interference such as that developed for the *HMS2:BAT2* locus. To test this experimentally, we inserted a new transcription unit into *HMS2* to disrupt the *HMS2* and *SUT650* transcription units and asked what mutations would restore transcription between the beginning and end of these TUs.

## Engineering *HMS2* reveals the balance between sense and antisense states and demonstrates the plasticity of transcription units

The *HMS2* locus was engineered by insertion of the p*TEF.KanMX.TEF*t expression cassette (*Longtine et al., 1998*) at position +650 bp (*Figure 7A*). In this construct (*HMS2:TEF:Kan*), there are two promoters

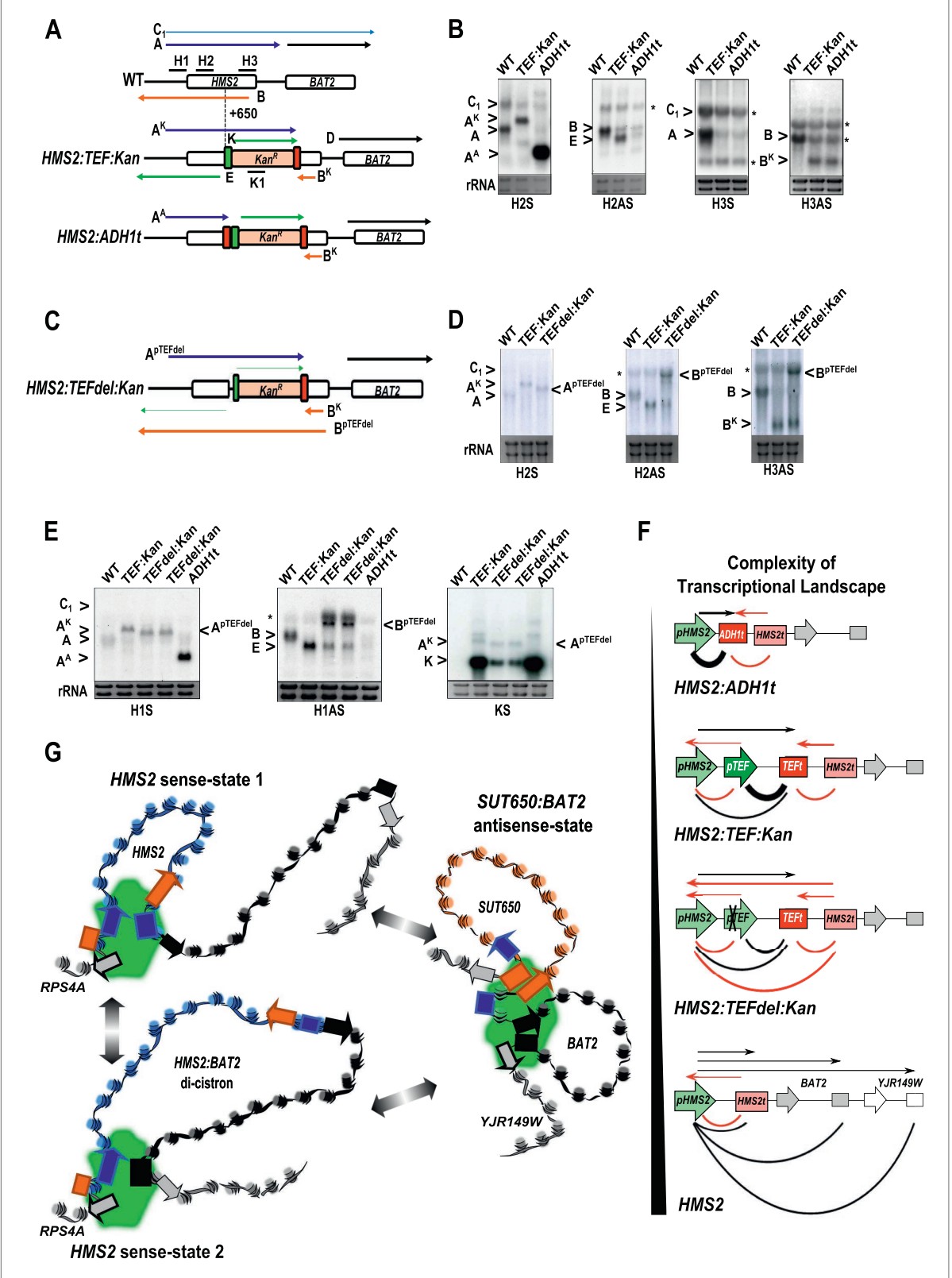

**Figure 7**. The plasticity of transcription units. (**A**, **C**) Schematic of constructs showing transcripts and position of probes. Transcripts resulting from the p*TEF:kanMX:TEFt* insertion, the same construct with the *ADH1* terminator (pink box), or the *TEF:kanMX:TEF* insertion with a deleted *TEF* promoter (p) are identified with a superscript K, A, or p*TEFdel*, respectively (see *Figure 7—figure supplement 1* for more details). (**B**, **D**, **E**) Northern blots in the strains

*Figure 7. Continued on next page*

*Figure 7. Continued*

indicated showing the conditional nature of the *TEF* terminator (t) when the p*TEF* is compromised. (**F**) Modelling transcription in an interleaved genome demonstrating an increase in complexity of the transcriptional landscape as the strength of terminators and promoters decreases. Constructs representing each state are indicated; grey and white boxes are downstream TUs. Solid directional boxes are promoters; squares are terminators. Deeper colour indicates increased strength. Arrows on top of diagrams represent the transcripts initiating or terminating at p*HMS2* and *HMS2*t (black sense; red antisense). Curved lines under diagrams show the extent of each individual transcription unit. Transcription units are envisaged as dynamic with any one region only involved in one TU at any time in an individual cell (see *Figure 7G*). Weakening of p*TEF* in the *HMS2:TEF:Kan* construct results in a level of transcription complexity (number of different TUs) similar to the native *HMS2* locus. (**G**) Schematic showing the dynamic alternative transcription units at the *HMS2:SUT650:BAT2* locus. Directional boxes are promoters, squares are terminators coloured coded according to the nature of the TU to which they belong (blue, *HMS2*; black, *BAT2*; Orange, *SUT650*; Grey, *RPS4A* and *YJR149W*). *HMS2* forms alternative TUs with its own terminator (sense-state 1) or with the *BAT2* terminator (sense-state 2—a di-cistronic transcription unit) excluding the *SUT650*, *BAT2*, and *YJR149W* promoters from a region where transcription can occur (green shading). *RPS4A*, an OX gene expressed divergently from *HMS2* is active. The sense state predominates in the OX phase of the YMC and during growth on GLU. Any promoter or terminator excluded from the transcription machinery is neutral to the transcription process, explaining how both promoters and terminators are subject to extensive read-through transcription. The sense-state toggles to the alternative antisense-state, predominant in GAL and during the RC phase of the YMC. Here, the formation of the *SUT650* transcription unit excludes the *HMS2* and *RPS4A* promoters from the transcription machinery. This relieves interference of *BAT2* and *YJR149W* by *HMS2* sense transcription. One transcription unit is envisaged to exist in one cell at any one time. The antisense state is the default, requiring transcription factor-dependent activation at *HMS2* to switch to the sense state.

The following figure supplement is available for figure 7:

**Figure supplement 1**. Creating and characterizing an interleaved locus around *HMS2*.

in tandem (*HMS2* and *TEF*) and two terminators in tandem (*TEF* and *HMS2*). This is sufficient to disrupt the *HMS2* and *SUT650* transcription units. The *SUT650* antisense promoter is functional and produces a truncated antisense transcript (B[K]) that ends in the vicinity of *TEF*t (*Figure 7A,B*). Thus *TEF*t functions bi-directionally to terminate *SUT650* (B[K]), the sense transcript from p*TEF*, expressing the kanamycin resistance (*KanMX*) transcript (K), and the sense transcript (A[K]) initiated at the *HMS2* promoter and terminating at *TEF*t (*Figure 7A,B*). p*TEF* is a well-characterised (*Steiner and Philippsen, 1994*) strong bi-directional promoter and produces an antisense transcript (E) extending into p*HMS2*. To confirm this, we inserted *ADH1*t directly upstream of p*TEF* (*Figure 7A*). This resulted in increased levels of a truncated sense transcript (A[A]) and considerably reduced levels of antisense transcript (E) (*Figure 7B*).

We reasoned that by introducing mutations into the *TEF* promoter, we would be able to ask how disabling p*TEF* affects the *HMS2* and *SUT650* transcription units. Using the *HMS2:TEF:Kan* construct as a starting point, mutations were introduced into the *TEF* promoter, including deletions of putative transcription factor binding sites (*Figure 7C* and *Figure 7—figure supplement 1*). A long deletion (*HMS2.TEFdel1:Kan*), removing all the *TEF* promoter sequences upstream of the TATA box, results in a significant reduction in expression of the *KanMX* cassette itself (transcript K) and in levels of the antisense transcript (E) (*Figure 7D,E* and *Figure 7—figure supplement 1*). Remarkably, in this strain, we observed a new antisense transcript from the 3′ end of *HMS2* that reads through *TEF*t (B[pTEFdel]) (*Figure 7C–E*). We confirmed that the B[pTEFdel] transcript spans the 3′ region of *HMS2* using the H3AS probe (*Figure 7D*). Thus, the disabled weak p*TEFdel1* resulted in reduced functionality at *TEF*t allowing B[pTEFdel], initiating at the *SUT650* antisense promoter, to read through *TEF*t, the *KanMX* cassette and into the *HMS2* promoter (*Figure 7C–E*). This experiment illustrates (i) the conditional nature of the bi-directional *TEF* terminator and (ii) the remarkable plasticity of interleaved transcription units. We suggest that the p*TEF.KanMX.TEF*t expression cassette is strongly expressed (compare levels of transcripts A[K] and K in *Figure 7E*), effectively preventing the formation of the *HMS2:SUT650* transcription units. When the *TEF* promoter is disabled, reducing expression of the p*TEF.KanMX.TEF*t expression cassette, the *SUT650* antisense transcription unit can form, resulting in the appearance of the long antisense transcript. Alternatively, reduced *KanMX* sense transcription decreases the ability of a terminator to stop antisense transcription from the *SUT650* antisense promoter. This illustrates the complex relationships that control levels of expression in the interleaved yeast genome (*Figure 7F*).

## Discussion

We propose a two-state model for transcription at *HMS2* and *BAT2* that combines the effects of temporal separation, overlapping interfering transcription able to bypass typical termination and

promoter signals, and insulation from interference by the formation of an antisense transcription unit (*Figure 7G*). There are two states: a sense-dominant state (blue transcripts) and an antisense-dominant state (orange transcripts) with respect to *HMS2* (see *Figures 6B and 7G*). Transcription factors (Input) at the *HMS2* promoter regulate sense transcription and loss of input results in the antisense-dominant state. *SUT650* transcription insulates *BAT2* from interference by *HMS2*, reinforcing the antisense dominant state-switch. We propose that transcription in the *HMS2* sense orientation transmits the state-defining regulatory signals to neighbouring genes through mechanisms of *cis*-acting transcriptional interference, thus repressing *BAT2*, *YJR149W,* and *SUT650*. By these means, reciprocal regulation in cycles or on environmental change could be achieved.

More generally, how far a transcription event extends into the regulatory sequences of its SAP, or the regulatory sequences of an upstream or downstream gene may determine whether that region is competent to make a state-switch or not and respond to environmental or metabolic cues. Our genome-wide analysis supports many more examples of the type of organisation we observe at *HMS2:BAT2*. Moreover, the *HMS2:BAT2* cluster is conserved in a range of yeast strains expected of a functional linkage. There are many loci at which di-cistronic read-through transcripts are evident (*Pelechano et al., 2013*), raising the possibility that transcriptional interference leading to reciprocal expression of genes in tandem arrays is a much more widespread phenomenon in yeast.

A di-cistronic transcript means that RNA polymerase has to read through the region determining polyadenylation and termination (the terminator). Moreover, antisense transcription arising from divergent promoters may have to bypass the terminator of the upstream gene. This work indicates the capacity of a terminator to function is conditional on the strength of the proximal promoter. We suggest that this reflects the role for promoters in defining the beginnings and ends of TUs (*El Kaderi et al., 2009*; *O'Sullivan et al., 2004*); stronger promoters preferentially selecting the proximal terminator.

The role of Sua7 in preventing read-through transcription is intriguing given its association with the 3′ region of genes, polyadenylation/transcription termination, and higher order structures in chromatin (*Medler et al., 2011*). This raises the possibility of an antisense transcription unit-specific function for TFIIB at the 3′ ends of genes, characterised here using the *sua7-1* allele. A simple scenario in which 3′end TFIIB directs transcription by promoting both antisense transcription and physically terminating sense transcription is compatible with the interference/insulation model proposed here. It is also plausible that alternative higher order structures linking the sense or antisense promoters to one of a possible number of terminators could also explain dynamic state-switching between sense and antisense or multi-cistronic transcripts (*Murray et al., 2012*) (*Figure 7G*). Only one structure would exist in each cell at any one time leading to temporal separation of the different TUs. By this model only the juxtaposed regions would function in transcription initiation and termination, allowing other promoters or terminators between these regions to be ignored (*Figure 7F,G*). In either case, our data support the idea that the functions of these linked regulatory elements are coupled through the act of transcription. Thus, the interleaved architecture of the yeast genome not only lends itself to high degrees of transcriptional overlap, but more importantly, to an equally high degree of transcription unit interdependency by co-opting transcription itself as a regulatory mechanism.

While this study has been predominantly conducted using a single locus approach in yeast, the implications of this work could be far reaching. For instance, establishing wider, biologically relevant transcriptional landscapes through sense/antisense state-switching could be pertinent to aspects of chronobiology such as circadian rhythms (*Kramer et al., 2003*; *Feng and Lazar, 2012*), the mitotic/meiotic cell cycles (*Morris and Vogt, 2010*), and other developmental processes in higher eukaryotes. This work places the sense/antisense relationships in a broader biological context and sheds light on how transcription might work as the nexus in more extensive networks that are organized in both spatial and temporal dimensions.

## Materials and methods

All experiments were performed at least in duplicate to ensure that the trends observed were reproducible. Briefly, unless otherwise stated, cells were grown to exponential phase in rich medium (YP) supplemented with 2% glucose (YPD) or 2% galactose (YPGal). The strains of *S. cerevisiae* used for this study are shown in *Supplementary file 1L*. The Anchor Away technique exploits the high affinity interaction between the FK506 Binding Protein (*FKBP12*) and *FKBP12* Rapamycin binding (FRB) domain of human mTOR protein in order to deplete target proteins from the nucleus in a time-dependent manner (*Haruki et al., 2008*). Control and treated cells were harvested after 1 h of DMSO

or rapamycin exposure. Total RNA extracts were acquired by hot phenol extraction and enriched for polyadenylated transcripts using Oligotex-dT as directed by the manufacturer (QIAGEN NV.). The concentration of RNA extract was measured and standardised to 1 µg/µl using a nanodrop spectrophotometer. The protocol for rapid amplification of cDNA 3' ends (3' RACE) used in this study was modified from the methods provided by the 3' RACE System for Rapid Amplification of cDNA Ends kit (Invitrogen Carlsbad, CA). Primers used are shown in *Supplementary file 1M*. Northern blotting was done as described in *Murray et al. (2012)*. In vitro transcription with T7 RNA polymerase and radiolabelling was used to create strand-specific probes for Northern blotting. Primers are shown in *Supplementary file 1N*. Hybridization of RNA to strand-specific, high resolution tiling arrays was performed as described in *Perocchi et al. (2007)*. Data analysis was performed by members of the Lars Steinmetz group at the European Molecular Biology Labs in Heidelberg, Germany (http://steinmetzlab.embl.de/cgi-bin/viewMellorLabArray.pl?showSamples=502_Glu1Vs503_Gal1&type=heatmap&gene=hms2). Native elongating transcript sequencing, (NET-seq) was done and analysed as described in *Churchman and Weissman (2011)*. To assess the genome-wide significance of our observations, all genes in *Supplementary file 1A* were characterised according to their orientation with respect to their upstream (5P divergent, 5P tandem) or their downstream gene (3P, convergent), their own, and their neighbouring genes' gene type (ORF, CUT, SUT, other) and YMC status (OX, oxidative; RB, reductive building; RC, reductive charging; NA/NC, non-cycling) and the number of genes in each category and median expression level in glucose and galactose determined. Genes that were located more than 1,000 bp from their neighbouring genes were excluded from the analysis. Computer simulations were performed by shuffling the strand location, gene type, YMC status, and expression levels of individual genes on every chromosome while keeping distances between genes and the total number of genes on each strand and of each type constant (*Source code 1*–MATLAB codes). Expression levels were simulated by randomly selecting recorded expression levels. After each simulation run, the number of genes and the median expression levels were calculated for each category. Simulations were run 10,000 times. Overlapping genes were defined as being positioned on the opposite strand and having 3' ends covering the same region of the genome. Enrichment analysis for factors was performed using the published data set by *Venters et al. (2011)*. The dual-labelled single molecule RNA Fluorescence *In Situ* hybridization (FISH) protocol was adapted from *Zenklusen and Singer (2010)*. Fluorescently labelled DNA probes to detect sense or antisense transcripts are shown in *Supplementary file 1O*. Image acquisition was performed with a DeltaVision CORE: Wide-field fluorescence deconvolution imaging microscope. Using a 100x objective lens, 31 'z-stacks' were collected. For each stack, an exposure time of 0.5 s for DAPI and 1 s for Cy3 (TRITC) and Cy5 filters was applied. Images were captured using a CoolSNAP HQ camera (Photometrics, Tucson, AZ). To facilitate image analysis, 3D data sets were compressed into 2D images by using a maximum projection, using *hms2Δ* cells to determine threshold signal for expression. Images displayed as 2D compressions of z-stacks 12–22 (11 stacks), which included most of the nucleus and cytoplasm. Error bars represent standard error of the mean. Chromatin immunoprecipitation (ChIP) was performed as described by *Morillon et al. (2003, 2005)*. Primers used for RT qPCR are displayed in *Supplementary file 1P*.

## Acknowledgements

The authors would like to thank Ilan Davis, Micron Oxford, and Dan Larsen for help with RNA FISH, Athar Ansari, and Mike Hampsey for strains containing the *sua7-1* allele and helpful advice, Stirling Churchman for help and advice with the NET-seq technique, Anitha Nair for excellent laboratory support, Benjamin Schuster-Böckler for advice regarding the computer simulations, and the following for funding; Keble College and the Clarendon Fund (to T.N.) and a Wellcome Trust Strategic Award (091911) supporting advanced microscopy at Micron Oxford (http://micronoxford.com)

## Additional information

### Competing interests

JM: Adviser to Oxford Biodynamics Ltd and Sibelius Ltd and sits on the board of Chronos Therapeutics. OBD provided funding for this work but like all the funders, had no say in the design or outcome of the research and do not benefit in any way from this research. The other authors declare that no competing interests exist.

## Funding

| Funder | Grant reference number | Author |
|---|---|---|
| Wellcome Trust | WT089156MA | Jane Mellor |
| National Institutes of Health | | Lars M Steinmetz |
| Oxford Biodynamics | ALRNEI1 | Jane Mellor |
| European Commission | Epigenesys FP7 Network | Jane Mellor |
| Deutsche Forschungsgemeinschaft | | Lars M Steinmetz |
| Natural Sciences and Engineering Research Council of Canada | | Tania Nguyen |
| Fundação para a Ciência e a Tecnologia | | Ana Serra Barros |
| Wellcome Trust | Graduate Studentships | Harry Fischl, Françoise S Howe, David Brown |
| Engineering and Physical Sciences Research Council | Graduate Studentships | Ronja Woloszczuk, Struan C Murray |

The funders had no role in study design, data collection and interpretation, or the decision to submit the work for publication.

## Author contributions

TN, FSH, RW, Conception and design, Acquisition of data, Analysis and interpretation of data, Drafting or revising the article; HF, ASB, ZX, DB, SCM, SH, JMH, LO'C, GS, Acquisition of data, Analysis and interpretation of data, Drafting or revising the article; LMS, JM, Conception and design, Analysis and interpretation of data, Drafting or revising the article

## Additional files

### Supplementary file

• Supplementary file 1. (**A**) Genome-wide NET-seq (NET) and poly(A)$^+$ RNA hybridised to microarray (mi). (**B**) Data for Pie Charts in *Figure 1*. (**C**) YMC genes (Coloured in columns A, C, and E) that overlap with genes whose transcription changes >threefold on the GLU to GAL shift (Column G—complete list colour coded). (**D**) Gene Ontology (GO) associated with genes that change >threefold on the GLU to GAL shift. (**E**) Annotated CUTs and SUTs that change >threefold on GLU to GAL shift (from Supplementary file 1A). (**F**) Extracted data from genome-wide simulation of gene type, orientation, and regulation. (**G**). Gene Groups from genome-wide annotations of OX.RC, RC.OX, and non-cycling (NC) pairs in tandem—used to provide information for Supplementary file 1J and to derive the distinct environments surrounding pairs of cycling or non-cycling genes. (**H**) Selected genes from Supplementary file 1A and analysis of their environment. (**I**) Genes that change >threefold on the GLU/GAL shift that also have an annotated antisense CUT or SUT that also changes >threefold on the GLU/GAL shift. (**J**) Selected gene clusters resembling *HMS2:BAT2*. (**K**) Transcription-related factors enriched at the promoters of RC or OX genes. (**L**) Genotype of yeast strains used in this study. (**M**) Primers used for 3'RACE. (**N**) Primers used to generate strand-specific probes for Northern blot analysis. The T7 promoter sequence is shown in parentheses. (**O**) RNA FISH probes. (**P**) Primers used for real-time PCR.

• Source Code 1. Source Codes in MATLAB 'simulation_gene_orientation_and_expression'.

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
