## [Decision Letter]

Thank you for sending your work entitled “Transcription mediated insulation and interference direct gene cluster expression switches during biological rhythms” for consideration at *eLife*. Your article has been favourably evaluated by James Manley (senior editor), a Reviewing editor, and 3 reviewers.

After a thorough discussion, the consensus among the reviewers and the editors was that this is an excellent paper reporting some remarkable findings, which would be worthy of publication in *eLife* after addressing the concerns listed below.

The metabolic cycle has emerged as a fundamental biological phenomenon that is critical for proper cell function. It has been appreciated that oxidative conditions are in conflict with reducing reactions, and vice versa. Consequently, cells have evolved mechanisms to temporarily separate these conflicting conditions, by oscillating both the oxygen consumption as well as the expression of ∼3000 genes. The remarkable precision of the synchrony in maintaining oscillating levels of mRNAs must involve intracellular events and communication, the basis of which remains obscure. This paper provides some solutions to this enigma by demonstrating how transcription of tandemly arrayed genes can regulate a bi-modal switch so that activation of one gene would repress the other and vice versa. This paper will serve the scientific community in various aspects. First, it will drive the attention of the readers to the complexity of the YMC, an issue that is still underrepresented in the literature. Second, it illustrates the critical roles played by transcription terminators in both terminating transcription and initiating transcription in the anti-sense direction. Third, transcription of anti-sense (AS) as a key modulator of gene expression, in general, and its involvement in the bi-modal switch, in particular. Fourth, and most importantly, the paper demonstrates, in many ways, the intricate relationship between transcription of sense, anti-sense and neighbouring genes in a manner that supports the bimodal switch.

The major concerns raised by the reviewers are:

1) Generality of findings. The impact of this work would certainly be higher upon defining the prevalence of the key regulatory mechanisms described. The paper focuses on two tandemly arrayed genes (HMS2/BAT2) whose expression switches during the YMC. They provide a detailed mechanism of the switching whereby transcription read-through into the downstream promoter downregulates transcription of the downstream gene and how AS transcription is involved. Their conclusion that the sense and AS transcription are in conflict, and that cells tend to separate the “sense state” from the “AS state” temporarily is by itself remarkable. However, how general is this mechanism? What fraction of YMC-regulated genes is adjacent or anti-sense? More than expected by chance? And then what fraction of adjacent or anti-sense transcripts is co-regulated in the YMC? Moreover, if the mechanism is more prevalent than just HMS2/BAT2, do the anti-sense transcripts always fall into opposite YMC categories, i.e., oxidative vs reductive? Is this locus even representative of other glu-gal clusters? If 10 loci are chosen at random (based on having one strong glu-gal-regulated transcript), do the neighbouring genes show potent regulation like the one well-explored region? Reviewers agreed that a more global assessment of the prevalence of the mechanism described would raise the impact significantly. However, it was noted that the bioinformatics involved for a global analysis would not be trivial. Thus, reviewers request that the authors manually mine the NET-seq data regarding transcript isoforms (like the data shown in Figure 2—figure supplement 1) and examine whether transcription of one gene invades the promoter of its neighbouring gene, and whether the AS transcription is anti-correlated. The control should be the same number of tandemly positioned gene pairs that are chosen at random. If the authors have the bioinformatic means to take a whole genome view, all the better.

2) Weak links to YMC and circadian rhythms. Although the reviewers appreciated the novelty and elegance of the molecular mechanisms discovered, there was agreement that in its current form the manuscript does not make a convincing argument about general relevance to YMC or circadian rhythms and it was agreed that the writing, including the title, should be tempered to focus more on the mechanistic aspects and less on the YMC and circadian biology. It is proposed that the title be changed to “Transcription mediated insulation and interference direct gene cluster expression switches”.

3) The choice of sua7-1 is not well communicated. The reader of the “Results” section is puzzled why TFIIB was chosen to begin, and why this specific allele was used, in particular. Only in the Discussion the reader learns that sua7-1 is defective in loop formation.

4) Figure 3 shows that the insertion of ADH1t abolished the signal detected by probe H2. From the point of view of the AS transcription, this probe is located downstream of ADH1t. The authors concluded that ADH1t blocks SUT650 AS transcription. However, it is possible that ADHt acts in both orientation and simply terminated transcription and therefore H2 could not detect any transcript. If the authors know that ADH1t does not act as terminator in the reverse orientation, they should indicate it. Moreover, even if the ADH1t is not acting as terminator of the AS, it can introduce a destabilization element that leads to its rapid degradation. In order to support this important conclusion, the authors could use a decay mutant strain (e.g., rrp6) and probe the membrane to detect the region between the ADHt and SUT650 promoter.

5) The replacement of HMS2 ORF with URA3 requires some clarifications. Why didn't the authors simply remove the region around the TSS, leaving all other sequences unchanged? Why did they prefer to introduce the entire URA3? There are two issues that need to be addressed. (1) The possibility that URA3 sequences, in both orientations, introduce stability biases onto the RNAs. (2) The possibility that the URA3 sequence contains a terminator in the AS direction, explaining why the AS RNA was not detected by H1 probe, located downstream of this hypothetical terminator. These issues can be discussed or addressed experimentally.

6) In Figure 6—figure supplement 1, the authors used TFs with putative binding sites at intergenic region between HMS2 and BAT2 (no references were included). The membranes were probed only with the H1 probe. This case might be an opportunity to examine if any of these factors play a role in the transcription of BAT2, by probing the membrane with an appropriate probe. URA3 ORF can be replaced back to HMS2 ORF by homologous recombination and selection on 5-FOA, thus resulting in a precise deletion of just the region around the TSS. Maybe one of these TFs is required for BAT2 transcriptional repression by C1 transcription (e.g., transcription through the binding site might displace the TF).

---

## [Author Response]

*1) Generality of findings. The impact of this work would certainly be higher upon defining the prevalence of the key regulatory mechanisms described*.

We agree with the reviewers about this point and had, in fact, already embarked on a computational genome-wide analysis. Part of this very large dataset has been analysed for this work and is presented in the revised version of the paper.

*The paper focuses on two tandemly arrayed genes (HMS2/BAT2) whose expression switches during the YMC. They provide a detailed mechanism of the switching whereby transcription read-through into the downstream promoter downregulates transcription of the downstream gene and how AS transcription is involved. Their conclusion that the sense and AS transcription are in conflict, and that cells tend to separate the “sense state” from the "AS state" temporarily is by itself remarkable. However, how general is this mechanism*?

From the point of switching sense and antisense states there is now one experiment in a recent paper from the Zenklusen lab supporting the idea of separation of the sense and antisense states in individual cells, in this case at *PHO84* (Castelnuovo M, Rahman S, Guffanti E, Infantino V, Stutz F and Zenklusen D: Bimodal expression of PHO84 is modulated by early termination of antisense transcription. Nat Struct Mol Biol 20: 851-8). We also have as yet unpublished evidence that at *GAL10*, the sense and antisense transcripts are present in different cells during early induction. Given the FISH evidence for YMC regulated genes from the Singer and Botstein labs, we think that switching is likely to be common.

What fraction of YMC-regulated genes is adjacent or anti-sense? More than expected by chance? And then what fraction of adjacent or anti-sense transcripts is co-regulated in the YMC?

We have data in the revised manuscript addressing these issues (Tables S5 to S9). The antisense question is hard to address adequately due to the relatively poor annotation of the antisense transcriptome – being limited to CUTs and SUTs. There are 1772 annotated CUTs and SUTs but much more transcription antisense to genes is evident from the NET-seq analysis. A separate analysis suggests that 75% of all yeast genes are subject to some antisense transcription in the vicinity of the sense promoter that influences the chromatin organization around that promoter. Nevertheless, our analysis for this work suggests that YMC genes are much more likely to have antisense transcription, although the number of genes is relatively small because of the high thresholds (>3-fold change in GLU/GAL both the YMC gene and the SUT or CUT). 71 of the 206 genes (34.4%) with regulated sense and antisense have SUTs/CUTs whose transcription also changes 3-fold in the GLU/GAL shift. 26% of all SUTs and CUTs show a >3-fold change on the GLU/GAL shift. Thus regulated SUTs and CUTs are enriched opposite to ORFs that also show a > 3-fold shift.

*Moreover, if the mechanism is more prevalent than just HMS2/BAT2, do the anti-sense transcripts always fall into opposite YMC categories, i.e., oxidative vs reductive*?

Our selected gene study shows that for the 10 OX.RC tandem genes (like *HMS2.BAT2*), 9 (90%) of the OX genes have an antisense (annotated or not) that shows reciprocal transcription to the OX gene.

*Is this locus even representative of other glu-gal clusters*?

Yes, see Table S9 and Figure 1—figure supplement 1, Figure 1—figure supplement 2, Figure 1—figure supplement 3 and Figure 1—figure supplement 4.

*If 10 loci are chosen at random (based on having one strong glu-gal-regulated transcript), do the neighbouring genes show potent regulation like the one well-explored region*?

Yes, see Table S9 and Figure 1—figure supplement 1, Figure 1—figure supplement 2, Figure 1—figure supplement 3 and Figure 1—figure supplement 4.

*Reviewers agreed that a more global assessment of the prevalence of the mechanism described would raise the impact significantly*.

I hope that we have persuaded the reviewers that this is likely to be a more general phenomenon.

*However, it was noted that the bioinformatics involved for a global analysis would not be trivial. Thus, reviewers request that the authors manually mine the NET-seq data regarding transcript isoforms (like the data shown in*
Figure 2—figure supplement 1*) and examine whether transcription of one gene invades the promoter of its neighbouring gene, and whether the AS transcription is anti-correlated. The control should be the same number of tandemly positioned gene pairs that are chosen at random. If the authors have the bioinformatic means to take a whole genome view, all the better*.

We have used both approaches to provide the data in order to conclude this is a wider phenomenon.

*2) Weak links to YMC and circadian rhythms. Although the reviewers appreciated the novelty and elegance of the molecular mechanisms discovered, there was agreement that in its current form the manuscript does not make a convincing argument about general relevance to YMC or circadian rhythms and it was agreed that the writing, including the title, should be tempered to focus more on the mechanistic aspects and less on the YMC and circadian biology. It is proposed that the title be changed to “Transcription mediated insulation and interference direct gene cluster expression switches”*.

We are happy to do this and have removed this from the title and tempered our references to these mechanisms in the text.

*3) The choice of sua7-1 is not well communicated. The reader of the “Results”" section is puzzled why TFIIB was chosen to begin, and why this specific allele was used, in particular. Only in the Discussion the reader learns that sua7-1 is defective in loop formation*.

We agree with this point and have included more details in the Results section.

*4)*
Figure 3
*shows that the insertion of ADH1t abolished the signal detected by probe H2*.

This is correct for the H2AS probe.

*From the point of view of the AS transcription, this probe is located downstream of ADH1t*.

This is correct.

*The authors concluded that ADH1t blocks SUT650 AS transcription. However, it is possible that ADHt acts in both orientation and simply terminated transcription and therefore H2 could not detect any transcript*.

This is correct.

*If the authors know that ADH1t does not act as terminator in the reverse orientation, they should indicate it*.

We do know that the ADH1t acts as a terminator in both directions. It does so in its native context (terminating the YOL086AW TU) and when used out of context in *GAL10* (see [33] NAR).

*Moreover, even if the ADH1t is not acting as terminator of the AS, it can introduce a destabilization element that leads to its rapid degradation. In order to support this important conclusion, the authors could use a decay mutant strain (e.g., rrp6) and probe the membrane to detect the region between the ADHt and SUT650 promoter*.

We have included two additional pieces of data in Figure 3 to address this. First we have data showing the effect of the decay mutant strain (*rrp6*) using probe H2AS. There are no unstable transcripts detectable in the *rrp6* strain.

Second we use probe H3AS to show that SUT650 is prematurely terminated. Thus both *HMS2* sense transcription and *SUT650* antisense transcription are prematurely terminated in this strain.

*5) The replacement of HMS2 ORF with URA3 requires some clarifications. Why didn't the authors simply remove the region around the TSS, leaving all other sequences unchanged? Why did they prefer to introduce the entire URA3*?

We chose to completely replace the *HMS2* ORF with the *URA3* ORF so as to avoid problems with changing codons, deleting bases, or altering transcript stability while leaving the *HMS2* flanking sequences unaltered.

*There are two issues that need to be addressed. (1) The possibility that URA3 sequences, in both orientations, introduce stability biases onto the RNAs. (2) The possibility that the URA3 sequence contains a terminator in the AS direction, explaining why the AS RNA was not detected by H1 probe, located downstream of this hypothetical terminator. These issues can be discussed or addressed experimentally*.

We cannot address this using experimental data as BY4741 is *ura3Δ* and thus the *URA3* gene is not in our datasets to test for the presence of an early terminator. Given that the promoter of a gene is reported to determine sense RNA stability (Bregman A, Avraham-Kelbert M, Barkai O, Duek L, Guterman A and Choder M: Promoter elements regulate cytoplasmic mRNA decay. Cell 147: 1473-83; Trcek T, Larson DR, Moldon A, Query CC and Singer RH: Single-molecule mRNA decay measurements reveal promoter- regulated mRNA stability in yeast. Cell 147: 1484-97), we felt the *URA3* ORF replacement was an appropriate strategy. We have discussed these potential problems in the text and how we have interpreted our data given these provisos.

*6) In*
Figure 6—figure supplement 1*, the authors used TFs with putative binding sites at intergenic region between HMS2 and BAT2 (no references were included). The membranes were probed only with the H1 probe. This case might be an opportunity to examine if any of these factors play a role in the transcription of BAT2, by probing the membrane with an appropriate probe. URA3 ORF can be replaced back to HMS2 ORF by homologous recombination and selection on 5-FOA, thus resulting in a precise deletion of just the region around the TSS. Maybe one of these TFs is required for BAT2 transcriptional repression by C1 transcription (e.g., transcription through the binding site might displace the TF)*.

We apologise for not including references to the putative TF binding sites – this is now done. Putative factor binding sites, and even ChIP data with tagged factors, does not provide evidence that a particular factor is functional at a particular site, for a variety of reasons (Lenstra TL, Benschop JJ, Kim T, Schulze JM, Brabers NA, Margaritis T, van de Pasch LA, van Heesch SA, Brok MO, Groot Koerkamp MJ, Ko CW, van Leenen D, Sameith K, van Hooff SR, Lijnzaad P, Kemmeren P, Hentrich T, Kobor MS, Buratowski S and Holstege FC: The specificity and topology of chromatin interaction pathways in yeast. Mol Cell 42: 536-49, 2011). Indeed, many of the experiments linking transcription factors to changes in particular transcriptional responses at genes are likely to be artefacts resulting simply from altered growth rate as a result of the genetic intervention (O'Duibhir E, Lijnzaad P, Benschop JJ, Lenstra TL, Leenen D van, Groot Koerkamp MJA, Margaritis T, Brok MO, Kemmeren P and FCP H: Cell cycle population effects in perturbation studies. Mol Syst Biol. 10: 2014; Slavov N, Airoldi EM, van Oudenaarden A and Botstein D: A conserved cell growth cycle can account for the environmental stress responses of divergent eukaryotes. Mol Biol Cell 23: 1986-97, 2013), (and the different times spent in the OX and RC phases of the YMC). We think this criticism applies to all the experiments we have done with TF KO strains in Figure 6 and Figure 6—figure supplement 1 and have been careful not to over-interpret the data to infer that these TFs actually bind to the DNA. Rather, the data in Figure 6 showing the switching of the balance of *HMS2* sense and *SUT650* antisense suggest that ablation of some factors coupled to mild nutrient limitation leads to cells where transcription in the *HMS2:SUT650* region is predominantly in sense or antisense orientations. This provides evidence for reciprocal switching between *HMS2* sense and *SUT650* antisense and suggests a signalling input regulates this only at the *HMS2* promoter. We have a number of experiments in the paper to support this, particularly those in Figure 5. In summary these experiments show that expression of the antisense state (*SUT650* and *BAT2*) is constitutive in the absence of sense transcription, regardless of growth conditions (GLU or GAL). Normally *SUT650* transcription increases on the GLU to GAL shift but this reflects the reduction in *HMS2* sense transcription in GAL. Without sense, SUT650 levels remain high in GLU. Similar arguments apply to *BAT2*. Elimination of sense transcript in the p*GAL1:HMS2* anchor away experiment in Figure 5, by addition of Rapamycin to deplete nuclear Gal4 from cells in GAL, restores *BAT2* transcript levels. Thus the primary *BAT2* regulation is transcriptional interference by the sense transcription in GAL (in this experiment only as the inserted *GAL1* promoter is GAL-regulated).

Our view is that the (semi-) quiescent RC phase in the YMC is likely to be the default and RC genes would be expressed unless a specific growth signal is received. *HMS2* switches on and interferes with the transcription of *SUT650* and *BAT2* as long as the growth signal remains. If there is no growth signal, transcription of *BAT2* and *SUT650* is constitutive, as if cells are in the quiescent state that occurs naturally when nutrients are limited. In fact all the changes that occur when yeast naturally enter stationary phase due to nutrient depletion also occur in RC cells (Shi L, Sutter BM, Ye X and Tu BP: Trehalose is a key determinant of the quiescent metabolic state that fuels cell cycle progression upon return to growth. Mol Biol Cell 21: 1982-90). This does not rule out TFs in the regulation of RC genes, but it does suggest they are not condition-specific. This is supported by our two new genome-wide analyses now included in the manuscript (Table S10). We have examined the nature of the transcription factors’ association with the promoters of RC or OX genes (in GLU) from the Venters and Pugh work (2011) (now included in reference list). They looked at the association of 202 factors at yeast promoters. We used their dataset to ask whether OX and RC genes are enriched for particular factors, as both will be expressed in GLU-grown cells. We show significantly different (p<0.01) ChIP-Chip binding patterns for factors at the two groups of genes. The RC genes are enriched for cohesion, the CCR4/NOT complex, elongator, chromatin and chromatin remodelling factors and four TFs, (Fhl1, Ino4, Ume6 and Skn7). By contrast, OX genes are associated with GTFs, TEFs, TFIID, and the TFs Ifh1 and Rap1. Although “stress” related transcription factors may play a role at the RC genes, these data suggest chromatin-mediated regulation as a central feature for RC genes. Our data support TI-mediated chromatin changes at the *BAT2* promoter.

So to the question of whether the putative TF binding sites between *HMS2* and *BAT2* regulate *BAT2* expression or more simply, contribute to limiting *HMS2* read-through (Figure 6—figure supplement 1). As it stands, our data, albeit quite crude, provides no evidence for a role for these factors in activation of either *HMS2* or *SUT650*. We rule out TFs as roadblocks (at least Gal4 for *SUT650* transcribed into the *GAL* promoter) but do see a role for Gln3 (only in minimal medium) in preventing transcript C (*HMS2:BAT2* read through) and a reduction in *SUT650* levels (supporting increased TI at the *HMS2:BAT2* intergenic region). Given the literature supports a role for TFs and promoters in determining transcript stability, another explanation for the effect of loss of Gln3 is that transcript C is stabilised, but there is no change in transcription *per se* and thus no change in the transcript ratios. How RC genes are regulated is clearly complicated and beyond the scope of what we can do for a 2 month revision. We think the suggestion of the referee is excellent but have a number of reservations about this type of approach, especially as some of the putative TF binding sites are within the *HMS2* ORF (similar arguments to the *URA3* question above). Instead we have provided an Anchor-Away experiment to deplete nuclear Gln3 in Galactose (conditions expected to increase *BAT2* and *SUT650* expression). In the same experiment nuclear depletion of Sua7 results in loss of both *BAT2* and *SUT650* transcripts. We find no evidence that Gln3 is required for *BAT2* or *SUT650* expression in these conditions and the Sua7 control suggests this is not simple a result of the *BAT2* transcript being particularly stable in GAL. These data are now included in Figure 6—figure supplement 1.